# Caffeine ingestion improves morning neuromuscular performance to evening levels in healthy females

**Akshay Singh, Stuart J. Hesketh**[ID]*

School of Medicine, University of Lancashire, Preston, United kingdom

* SHesketh5@lancashire.ac.uk

## Abstract

### Purpose

To investigate whether caffeine ingestion offsets the circadian-related decline in morning neuromuscular performance in females.

### Methods

Thirteen healthy females completed three experimental trials: i) evening placebo ($PM_{PLAC}$), ii) morning placebo ($AM_{PLAC}$), and iii) morning caffeine ingestion (6 mg·kg$^{-1}$; $AM_{CAFF}$). A within-subject, repeated-measures, crossover, single-blind placebo design was used, with all subjects serving as their own controls. Each trial consisted of a maximal voluntary isometric contraction (MVIC) of the knee extensors, followed by intermittent isometric contractions (6 s contraction, 4 s rest) until task failure, and a post-fatigue MVIC. Time to exhaustion, torque and peak force, surface electromyography (RMS amplitude, MDF frequency), rate of perceived exertion, and tympanic temperature were all recorded across conditions.

### Results

During $PM_{PLAC}$, peak force and time to exhaustion were significantly greater ($p < 0.05$) than for $AM_{PLAC}$. Peak force was higher ($p < 0.01$) in $AM_{CAFF}$ in both pre- (+33%) and post- (+45%) fatigue MVIC trials compared to $PM_{PLAC}$. Similarly, RMS was higher ($p < 0.01$) during both $AM_{CAFF}$ and $PM_{PLAC}$ compared to $AM_{PLAC}$, whereas MDF did not change ($p > 0.05$) across trials. $AM_{CAFF}$ also improved ($p < 0.01$) time to exhaustion by 43% compared to $AM_{PLAC}$, and significantly reduced ($p < 0.01$) rate of perceived exertion, but without any change ($p > 0.05$) in temperature.

### Conclusions

Morning ingestion of 6 mg·kg$^{-1}$ of caffeine effectively reverses diurnal reductions in neuromuscular performance of females by enhancing peak force and time to

**Data availability statement:** Data are available from DOI: https://doi.org/10.6084/m9.figshare.30382900.

**Funding:** The author(s) received no specific funding for this work.

**Competing interests:** The authors have declared that no competing interests exist.

exhaustion to evening levels. No change in MDF, but an increased RMS suggests a central rather than peripheral mechanism for caffeine's ergogenic effects.

## Introduction

Athletic and neuromuscular performance often exhibits diurnal variation, with many studies reporting lower outputs in the early morning and higher performance in the late afternoon or evening [1–4]. However, substantial inter-individual variability exists, influenced by factors such as chronotype and habitual activity patterns [5,6]. These patterns are well documented across strength [1,7], power [8], and endurance tasks [9,10], and reflects the broader influence of circadian regulation on human physiology. Core body temperature, heart rate, blood pressure, and hormonal secretion all fluctuate predictably across the day [11–13], creating time-of-day dependent physiological states that influence readiness for performance. Importantly, these fluctuations have practical implications; for example, athletes competing in morning events may be physiologically disadvantaged compared with later in the day, raising interest in strategies that can offset such declines in performance.

Circadian regulation produces distinct physiological states that influence exercise performance across the day. In the evening, elevated body temperature enhances metabolic reactions and muscle compliance [11,14], while faster nerve conduction and favourable hormonal profiles further facilitate strength, power, and endurance outputs [11,15]. Yet, experimental attempts to equalise temperature across time points demonstrate that thermal effects alone cannot fully explain diurnal variations in performance [16–19]. Instead, this variation likely arises from the coordinated influence of multiple systems, including neural drive, molecular changes in gene expression, and shifts in substrate metabolism [12,20]. Given this complexity, targeted investigation of neuromuscular properties provides a practical lens for understanding how circadian variation influences performance [15,21].

Caffeine is a well-established nutritional strategy with consistent ergogenic effects on exercise performance [22–24]. Acting as a non-selective adenosine receptor antagonist, it increases central nervous system excitability, augments neurotransmitter release, and reduces perceived exertion [25,26]. Meta-analyses confirm its efficacy across a wide range of tasks, with robust benefits for endurance performance and smaller but meaningful effects on maximal strength and power [27,28]. Doses of 3–6 mg·kg$^{-1}$ are most commonly reported to enhance both aerobic and anaerobic outputs, with peak plasma concentrations achieved within 45–60 minutes of ingestion [29,30]. Importantly, emerging evidence suggests that caffeine may counteract diurnal decrements in performance, making it a promising strategy to enhance morning neuromuscular function to that of evening levels [31,32], thus warranting further investigation.

Despite the general efficacy of caffeine, observed ergogenic effects are heterogeneous and task dependent [33]. For example, factors such as exercise modality, training status, dose, timing of ingestion, sex, habitual intake, and methodological factors such as blinding and expectancy [23,34,35] all play a role. Meta-analyses

report small-to-moderate average effects, but also highlight considerable between-study variability and a lack of trials directly addressing circadian timing [27,29]. These limitations restrict mechanistic inference and further impede the development of population-specific guidance. Sex-specific physiology further complicates interpretation and application. Caffeine metabolism is principally mediated by CYP1A2, whose activity is modulated by sex hormones and menstrual phase, with hormonal contraceptive use known to prolong caffeine clearance and alter plasma exposure, potentially influencing ergogenic efficacy [36,37]. Hormonal fluctuations also influence thermoregulation, neuromuscular efficiency, and substrate utilisation, all of which may interact with circadian timing to shape performance. Despite these considerations, most studies remain male-centric, with female participants underrepresented or insufficiently controlled for menstrual status, limiting the applicability of current recommendations [38].

While the ergogenic properties of caffeine are well documented, its capacity to counteract morning declines in neuromuscular performance, particularly in females, remains unclear. The present study therefore aims to determine whether acute caffeine ingestion (6 mg·kg$^{-1}$) can elevate morning neuromuscular performance to evening levels in healthy young females. Specifically, we sought to compare maximal voluntary isometric contraction, time to exhaustion, and indices of neuromuscular function assessed both before and following a fatiguing protocol, alongside electromyographic activity, and perceived exertion across morning placebo, morning caffeine, and evening placebo conditions. We hypothesised that (i) neuromuscular performance will be lower in the morning under placebo relative to the evening, (ii) morning caffeine will improve peak force, time to exhaustion, exertion ratings, and neuromuscular function under fatigued conditions compared with morning placebo, and (iii) increases in performance associated with morning caffeine ingestion will be accompanied by indices of central neural drive, not by alterations in thermoregulation.

## Methods

### Subjects

Thirteen healthy females volunteered to participate in this study (age 24.4 ± 1.1 years; body mass 59.0 ± 10.0 kg; height 159.5 ± 4.4 cm). All participants provided written informed consent prior to enrolment, recruitment took place between 10/05/25 and 30/07/25. The study was conducted in accordance with the Declaration of Helsinki, and received approval from the Ethics Committee of the School of Medicine, University of Lancashire (ethics code: 070.24.10.17), and complied with institutional standards for the collection, storage, and confidentiality of human data.

All participants were classified as light caffeine consumers (≤60 mg·day$^{-1}$), based on a validated caffeine consumption questionnaire [39]. Participants were recreationally active, engaging in regular physical activity 2–4 times per week, primarily consisting of aerobic exercise (e.g., running cycling, gym-based cardiovascular training) and recreational sport. None of the participants were involved in structured resistance or strength training programmes, and none had prior experience with isometric strength testing or laboratory-based neuromuscular assessments beyond the familiarisation procedures conducted as part of the study. To minimise confounding influences, participants were instructed to abstain from strenuous exercise outside of testing, as well as from caffeine, alcohol, and smoking for 48 h prior to the first experimental session and to maintain abstinence throughout the three consecutive testing sessions, given the known effect of nicotine on caffeine metabolism [40]. To reduce variability with hormonal fluctuations, all participants were tested between days 10–15 following the onset of menstruation, based on self-report of the first day of their most recent menstruation and typical cycle length, which were confirmed verbally at each laboratory visit. None of the participants were using hormonal contraceptives at the time of the study. Although diet and sleep were not objectively monitored, participants were instructed to refrain from food consumption for at least an hour before each lab visit and to report any irregular sleep patterns over the course of the study. To control for circadian influences, all trials for each participant were conducted at the same time-of-day. Participants were informed that the study involved both caffeine and placebo conditions, but were not informed of the specific condition administered at each session, consistent with the single-blind design.

Inclusion criteria were: female sex, age 18–35 years, recreationally active, free from known cardiovascular, neurological, or musculoskeletal disorders, classified as light caffeine consumers (≤60 mg.day⁻¹), self-reported regular menstrual cycles, tested between days 10–15 following menstruation, and not taking medications known to affect neuromuscular function, menstruation, or caffeine metabolism.

Exclusion criteria included pregnancy, current smoking, habitual high caffeine intake(>60 mg.day⁻¹), history of lower-limb injury within the preceding six months, diagnosed sleep disorders, use of oral contraceptives, history of menstrual irregularities or amenorrhea, or use of ergogenic supplements during the study period.

## Study design and supplementation

A within-subject, repeated-measures crossover design with a fixed condition order and single-blind placebo design was employed, with all participants serving as their own controls. The morning placebo ($AM_{PLAC}$) and evening placebo ($PM_{PLAC}$) conditions served as within-subject control phases against which the effects of caffeine ingestion and time-of-day were compared. Each participant attended the laboratory to complete three experimental sessions on three consecutive days in a fixed order: (i) evening placebo (17:00; $PM_{PLAC}$); (ii) morning placebo (08:00; $AM_{PLAC}$); and (iii) morning caffeine (08:00; $AM_{CAFF}$; 6 mg·kg⁻¹), as shown in Fig 1. This order was selected to accommodate the circadian design and to minimise potential caffeine carryover effects on placebo trials. The design enabled direct comparison of the main effects of

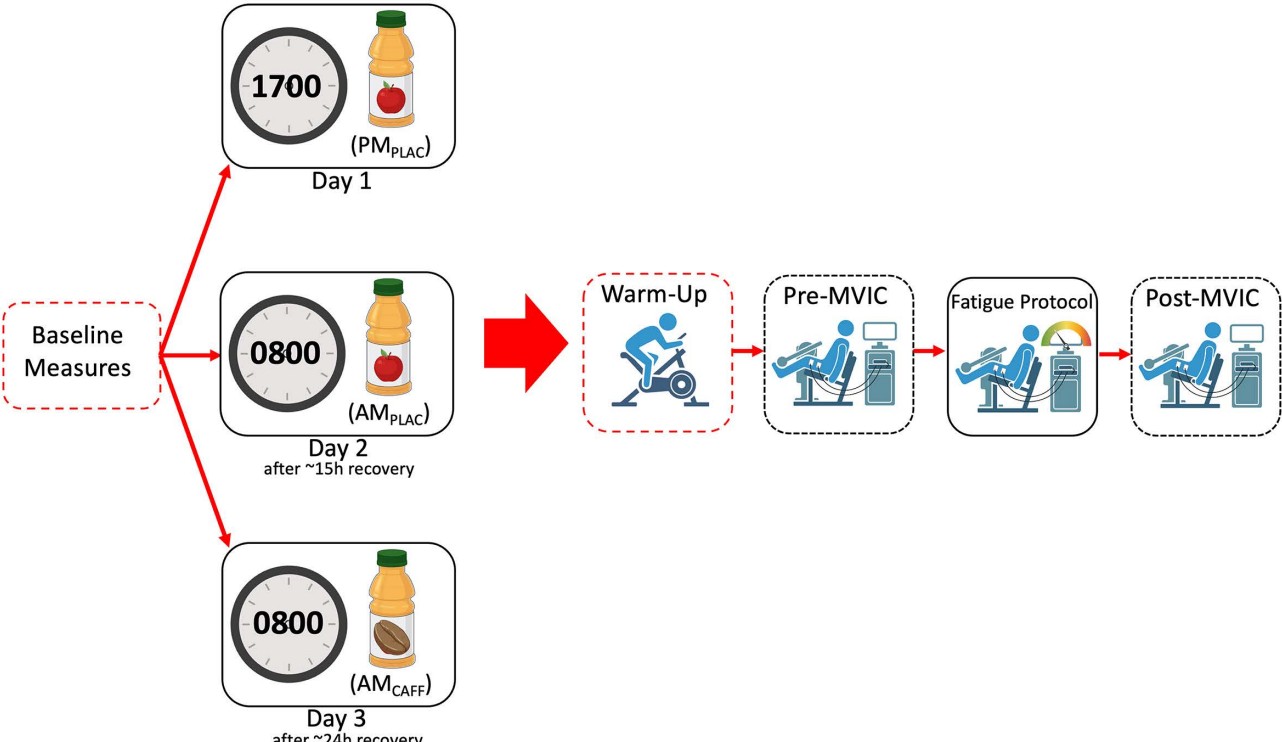

**Fig 1. Schematic of experimental design and procedures.** Thirteen healthy female participants completed three conditions in a repeated-measures, single-blind, placebo-controlled crossover design. Three trials were completed: morning placebo ($AM_{PLAC}$) at 0800, morning caffeine ($AM_{CAFF}$) at 0800 (6 mg·kg⁻¹), and evening placebo ($PM_{PLAC}$) at 1700. Anthropometric measures and core temperature were taken at the beginning of each trial. Each trial consisted of a 5 minute cycling warm-up at 90 rpm, 80 ± 10 W. The first (pre-) maximal voluntary isometric contraction (MVIC) was then performed for 5 seconds at 90 degrees knee flexion with 2 min rest periods between 3 efforts. A fatiguing protocol consisting of 6 second maximal isometric contractions interspersed by 4 seconds rest was then implemented, until peak torque declined to 50% of pre-MVIC levels, and then post-fatigue MVICs were performed. Surface electromyography and perceived exertion were measured throughout.

time-of-day (morning vs. evening) and caffeine ingestion (0 vs. 6 mg·kg⁻¹) on neuromuscular performance. This approach is consistent with established crossover designs in caffeine research, where treatment effects are determined through between-condition comparisons under matched conditions [41,42]. Testing times of 08:00 and 17:00 were selected based on prior evidence demonstrating consistent diurnal variation, with reduced morning (08:00) and enhanced evening (17:00) performance [4].

A caffeine dose of 6 mg·kg⁻¹ body mass was selected, as this dose is consistently associated with ergogenic benefits across endurance, strength, and power tasks and is known to elicit robust central nervous system effects in healthy adults [27,29]. For the caffeine condition ($AM_{CAFF}$), caffeine powder (ESN caffeine, Germany) was individually weighed for each participant and dissolved in a fruit smoothie consisting of 33 g carbohydrates, 1.6 g protein, 0.15 g fat, and 3.3 g fibre with a total of 143 kcal. In the placebo conditions ($AM_{PLAC}$ and $PM_{PLAC}$), the same fruit drink was provided without caffeine to ensure matched energy content and minimise likelihood of condition detection. Consistent with previous pharmacokinetic studies [43,44], participants ingested the caffeine drink 45 min before testing, expected to coincide with peak plasma concentrations, which typically occur 30–60 min post-ingestion [30,45], thereby maximising the likelihood of testing during peak bioavailability.

## Experimental protocol

All experimental sessions were conducted in July at the University of Lancashire, Preston, United Kingdom. This period provided a natural circadian window with sunrise at approximately 05:20 and sunset at 22:20. Outdoor temperature averaged 15.5±3.5 °C with relative humidity of 75±5%. Testing took place in a controlled laboratory environment where ambient temperature was maintained at 25±2 °C to minimise environmental influences on circadian-dependent performance [46].

Following Fig 1, baseline measurements were obtained at the start of each trial, including height (Seca 213 stadiometer, Seca GmbH & Co. KG., Germany), body mass (Seca GmbH & Co. KG., Hamburg, Germany), age, caffeine consumption [39], and tympanic temperature (Braun IRT 4520, Germany), a non-invasive surrogate of core temperature commonly used in exercise and circadian research [46]. Dominant leg was identified using a validated question-based method described in Van Melick et al, [47]. To reduce progressive learning effects, participants completed a familiarisation session during each laboratory visit, replicating the experimental protocol under submaximal conditions. Immediately following baseline measurements, to raise muscle temperature and prepare participants for exercise, all individuals performed a standardised warm-up consisting of 5 min of cycling on a stationary ergometer (Keiser M3i, Keiser Corporation, USA) at 90 rpm with a load of 80±10 W.

## Neuromuscular testing

Surface electromyography (EMG) of the vastus medialis was recorded throughout all testing procedures to measure neuromuscular activation. The vastus medialis was selected due to its primary role in knee extension, its superficial anatomy allowing reliable surface EMG recordings, and its frequent use as a representative quadriceps muscle in neuromuscular fatigue research [48]. Prior to electrode placement, the skin was shaved, lightly abraded, and cleaned with an alcohol swab to minimise impedance. Bipolar surface electrodes (Delsys Trigno™ SP-W06, 10 mm interelectrode distance) were positioned according to SENIAM guidelines [49], and the anatomical method described by Rainoldi et al, [48]. Electrodes were placed 4 cm above the superior border of the patella, oriented at a 55° angle to the line connecting the anterior superior iliac spine and the patella, aligned with muscle fibres and positioned away from the innervation zone. EMG was recorded from a single muscle to maximise signal quality, minimise cross-talk from adjacent muscles, and ensure repeatable electrode placement across experimental sessions [50]. The EMG signals were acquired in real-time using Trigno Discover software, with a band-pass filter of 20–450 Hz applied to minimise noise and motion artefacts before analysis. The EMG output from the vastus medialis was processed using EMGworks Analysis software (v.4.8.0; Delsys

Inc., Boston, MA, USA) and full wave rectified. Subsequently, the root mean square (RMS), representing the amplitude of muscle activation, and the median frequency (MDF), obtained via fast Fourier transform as an indicator of motor unit firing behaviour and fatigue, were calculated. A window length of 250 ms with 50% overlap was applied in accordance with established recommendations [51] to ensure adequate frequency resolution while minimising data dispersion.

Neuromuscular function was assessed using a HUMAC isokinetic dynamometer (HUMAC2015®, Version 15.000.0236; CSMi, Stoughton, MA, USA), calibrated before each session and adjusted to individual anthropometric dimensions. Participants were seated with the dominant leg secured to the lever arm, ensuring alignment of the lateral femoral epicondyle with the axis of rotation. The lower leg was fastened above the malleoli with a padded Velcro strap, and the arms were crossed over the chest to minimise extraneous movement. Positioning was standardised and replicated across sessions.

Maximal voluntary isometric contractions (MVICs) of the knee extensors were performed first (Fig 1). Each trial consisted of a 5-s maximal effort separated by 2 min rest intervals [22,52]. Three trials were performed, and the highest mean torque value was retained for analysis. Visual feedback and strong verbal encouragement were provided throughout to maximise effort. Following baseline MVICs, participants completed a fatiguing protocol adapted from Pethick and colleagues [52]. This consisted of repeated 6-s maximal isometric contractions interspersed with 4-s rest intervals, continued until torque output declined to 50% of baseline MVIC or volitional failure occurred. The number of completed contractions, time to exhaustion, and ratings of perceived exertion (RPE) were recorded. RPE were assessed using the Borg CR10 scale [53]. After a standardised 2-min rest period, participants performed post-fatigue MVICs to assess neuromuscular function under fatigued conditions.

## Experimental outcomes

The primary outcome of the study was maximal voluntary isometric contraction (MVIC) torque. Secondary outcomes included time to exhaustion, electromyographic variables (root mean square and median frequency), rating of perceived exertion, and core (tympanic) temperature.

## Statistical analysis

Normality was assessed using Shapiro-Wilk test and all data were found to be suitable for parametric testing. Data are reported as mean ± standard deviation (SD). All statistical analyses were conducted in GraphPad Prism (version 10.5.0; RRID:SCR_002798). One- or two-factor repeated-measures analysis of variance (ANOVA) tests were used to examine within-subject effects of condition and time-of-day. Greenhouse–Geisser corrections were applied, and significant main or interaction effects were followed by Tukey's post hoc comparisons. Statistical significance was accepted at $p < 0.05$. Effect sizes were reported using eta-squared ($\eta^2$) and partial eta-squared ($\eta p^2$). Peak torque values were normalised to body mass to facilitate comparison of neuromuscular performance between participants. BMI was not used as it does not account for body composition and is less informative for strength-related outcomes in healthy young adults. No a priori sample size calculation was performed for this study.

## Results

Thirteen healthy females completed all experimental trials. Participants were of similar anthropometric stature, 24.4 ± 1.1 years old, with a body mass of 59.0 ± 10.0 kg and height of 159.5 ± 4.4 cm. All were classified as light caffeine consumers (≤60 mg·day⁻¹) which was recorded using the caffeine consumption questionnaire [39]. No adverse events were reported, and all sessions were completed as scheduled. A summary of key outcome measures and statistics across conditions is presented in Table 1 with detailed comparisons below. Effect size analysis revealed moderate to large effects for condition across peak torque, time to exhaustion, RMS amplitude, and perceived exertion ($\eta^2 = 0.47–0.79$; Table 1), indicating that the observed differences were not only statistically significant but also of practical relevance.

**Table 1. Outcomes measures across experimental conditions.**

| Outcomes | PM Placebo (Mean ± SD) | AM Placebo (Mean ± SD) | AM Caffeine (Mean ± SD) | Main effect (ANOVA F, p, η²) | Significant pairwise differences (Tukey, p < 0.05) |
|---|---|---|---|---|---|
| Torque (N·m) Pre | 83.31 ± 12.34 | 67.23 ± 18.78 | 88.69 ± 21.23 | – | – |
| Torque (N·m/kg) Pre | 1.46 ± 0.34 | 1.14 ± 0.30 | 1.52 ± 0.26 | $F_{(1.763,\ 21.15)} = 10.77$, $p = 0.0009$, $\eta^2 = 0.47$ | $AM_{PLAC} < AM_{CAFF}$; $AM_{PLAC} < PM_{PLAC}$ |
| Torque (N·m) Post | 79.23 ± 17.24 | 62.69 ± 18.66 | 89.54 ± 24.39 | – | – |
| Torque (N·m/kg) Post | 1.38 ± 0.37 | 1.07 ± 0.33 | 1.55 ± 0.40 | $F_{(1.325,\ 15.90)} = 11.94$, $p = 0.0018$, $\eta^2 = 0.50$ | $AM_{PLAC} < AM_{CAFF}$; $AM_{PLAC} < PM_{PLAC}$ |
| TTE (s) | 82.2 ± 21.4 | 62.9 ± 18.0 | 89.9 ± 26.3 | $F_{(1.707,\ 20.48)} = 16.67$, $p < 0.0001$, $\eta^2 = 0.58$ | $AM_{PLAC} < AM_{CAFF}$; $AM_{PLAC} < PM_{PLAC}$ |
| RMS (µV × 10⁻⁵) Pre | 11.4 ± 5.59 | 8.28 ± 4.55 | 11.9 ± 6.72 | $F_{(1.77,\ 17.67)} = 11.96$, $p = 0.0007$, $\eta^2 = 0.54$ | $AM_{PLAC} < AM_{CAFF}$; $AM_{PLAC} < PM_{PLAC}$ |
| RMS (µV × 10⁻⁵) Post | 10.2 ± 5.25 | 7.80 ± 4.39 | 11.7 ± 7.16 | $F_{(1.38,\ 13.78)} = 10.18$, $p = 0.0039$, $\eta^2 = 0.50$ | $AM_{PLAC} < AM_{CAFF}$; $AM_{PLAC} < PM_{PLAC}$ |
| MDF (Hz) Pre | 68.05 ± 5.82 | 64.99 ± 5.38 | 68.52 ± 6.58 | $F_{(1.84,\ 18.38)} = 4.35$, $p = 0.031$, $\eta^2 = 0.30$ | $AM_{PLAC} < AM_{CAFF}$ |
| MDF (Hz) Post | 69.27 ± 5.01 | 66.44 ± 5.07 | 69.83 ± 5.84 | n.s. ($p = 0.1130$) | — |
| RPE | 6.31 ± 0.75 | 7.92 ± 0.49 | 6.23 ± 0.93 | $F_{(1.722,\ 20.66)} = 45.54$, $p < 0.0001$, $\eta^2 = 0.79$ | $AM_{PLAC} > AM_{CAFF}$; $AM_{PLAC} > PM_{PLAC}$ |
| Core Temp (°C) | 36.10 ± 0.64 | 35.32 ± 0.61 | 35.52 ± 0.66 | $F_{(1.854,\ 22.25)} = 10.95$, $p = 0.0006$, $\eta^2 = 0.48$ | $PM_{PLAC} > AM_{PLAC}$; $PM_{PLAC} > AM_{CAFF}$ |

Torque represents peak force normalised to body mass. Abbreviations are denoted: TTE, time to exhaustion; RMS, root mean square; MDF, median frequency.

## Torque

Maximal voluntary torque differed significantly across conditions. Pre-fatigue torque (Fig 2A) showed a main effect of condition, and post-hoc analysis revealed that torque was higher in $AM_{CAFF}$ (1.52 ± 0.26 N·m/kg) and $PM_{PLAC}$ (1.46 ± 0.34 N·m/kg) compared with $AM_{PLAC}$ (1.14 ± 0.30 N·m/kg), representing values of ~33% and ~28% respectively (both p < 0.01). A similar pattern was observed post-fatigue (Fig 2B), where post-hoc analysis indicated that torque in $AM_{CAFF}$ (1.55 ± 0.40 N·m/kg) and $PM_{PLAC}$ (1.38 ± 0.37 N·m/kg) was greater than in $AM_{PLAC}$ (1.07 ± 0.33 N·m/kg) with values ~45% and ~29% higher, respectively (both p < 0.01). No differences (p > 0.05) were detected between $AM_{CAFF}$ and $PM_{PLAC}$ at either time point. When pre- and post-fatigue values were analysed together (Fig 2C), torque in $AM_{PLAC}$ remained consistently lower than in both active conditions (all p < 0.01).

## Time to exhaustion

Time to exhaustion (Fig 2D) showed a main effect of condition, and post-hoc analysis showed it was longer in $AM_{CAFF}$ (89.9 ± 26.3 s) and $PM_{PLAC}$ (82.2 ± 21.4 s) compared with $AM_{PLAC}$ (62.9 ± 18.0 s), with values ~43% and ~31% higher, respectively (both p < 0.01). No differences (p > 0.05) were detected between $AM_{CAFF}$ and $PM_{PLAC}$.

## Electromyography

Pre-fatigue root mean square (RMS) (Fig 3A) indicated a main effect and differed significantly across conditions, post-hoc analysis revealed that $AM_{CAFF}$ (11.9 ± 6.7 x 10⁻⁵ µV) and $PM_{PLAC}$ (11.4 ± 5.6 x 10⁻⁵ µV) were higher than $AM_{PLAC}$ (8.28 ± 4.6 x 10⁻⁵ µV), with values ~44% and ~38% higher, respectively (both p < 0.01). Post-fatigue RMS (Fig 3B) followed the same

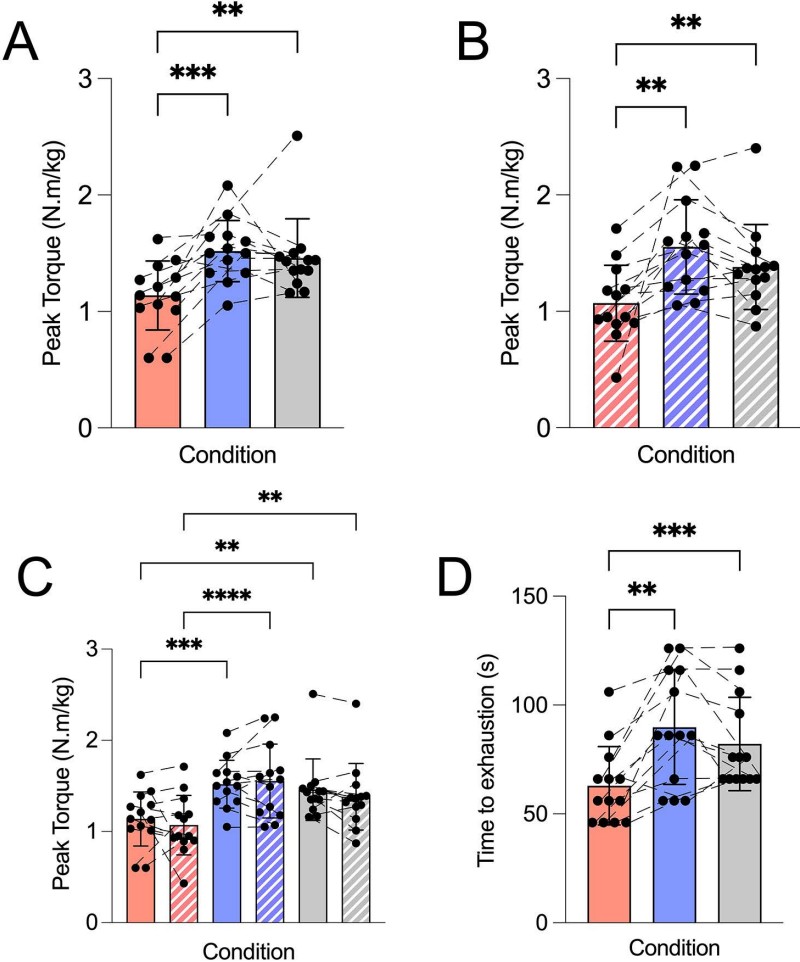

**Fig 2. Effect of time-of-day and caffeine ingestion on knee extensor isometric strength.** Morning caffeine (AM$_{CAFF}$) increased torque and time to exhaustion compared with morning placebo (AM$_{PLAC}$), reaching values comparable to the evening condition (PM$_{PLAC}$). Data are presented as mean±SD with individual values that represent single participants plotted. Trials were conducted in the morning (08:00) with placebo (AM$_{PLAC}$), with caffeine ingestion (6 mg.kg$^{-1}$; AM$_{CAFF}$), and in the evening (17:00; PM$_{PLAC}$). Red bars represent data for AM$_{PLAC}$ trial, blue bars for AM$_{CAFF}$, and grey bars for PM$_{PLAC}$. Solid bars indicate pre-fatigue MVIC, and hatched bars indicate post-fatigue MVIC. **(A)** Peak torque normalised to body mass (N·m/kg) during pre-fatigue MVIC. **(B)** Peak torque normalised to body mass (N·m/kg) during post-fatigue MVIC. **(C)** Comparison of pre- and post-fatigue MVIC torque. **(D)** Time to exhaustion in seconds, defined as the time until <50% of pre-MVIC or task failure. Statistical analysis was performed using one-way and two-way repeated-measures ANOVA with Tukey's post hoc tests. Significance is indicated as *p<0.05, **p<0.01, ***p<0.001, ****p<0.0001.

pattern, with AM$_{CAFF}$ (11.7±7.2 x 10$^{-5}$ µV) and PM$_{PLAC}$ (10.2±5.3 x 10$^{-5}$ µV) with values (7.80±4.4 x 10$^{-5}$ µV) ~50% and ~31%, higher than AM$_{PLAC}$, respectively (both p<0.01). No differences (p>0.05) were measured between AM$_{CAFF}$ and PM$_{PLAC}$ at either time point.

Pre-fatigue median frequency (MDF) (Fig 3D) showed a main effect of condition, and post-hoc analysis indicated that AM$_{CAFF}$ (68.5±6.6 Hz) was greater than AM$_{PLAC}$ (65.0±5.4 Hz), with values ~5% higher (p<0.05). PM$_{PLAC}$ (68.1±5.8 Hz) did not change (p>0.05) from either AM$_{CAFF}$ or AM$_{PLAC}$. Post-fatigue MDF (Fig 3E) also did not change across conditions (p>0.05).

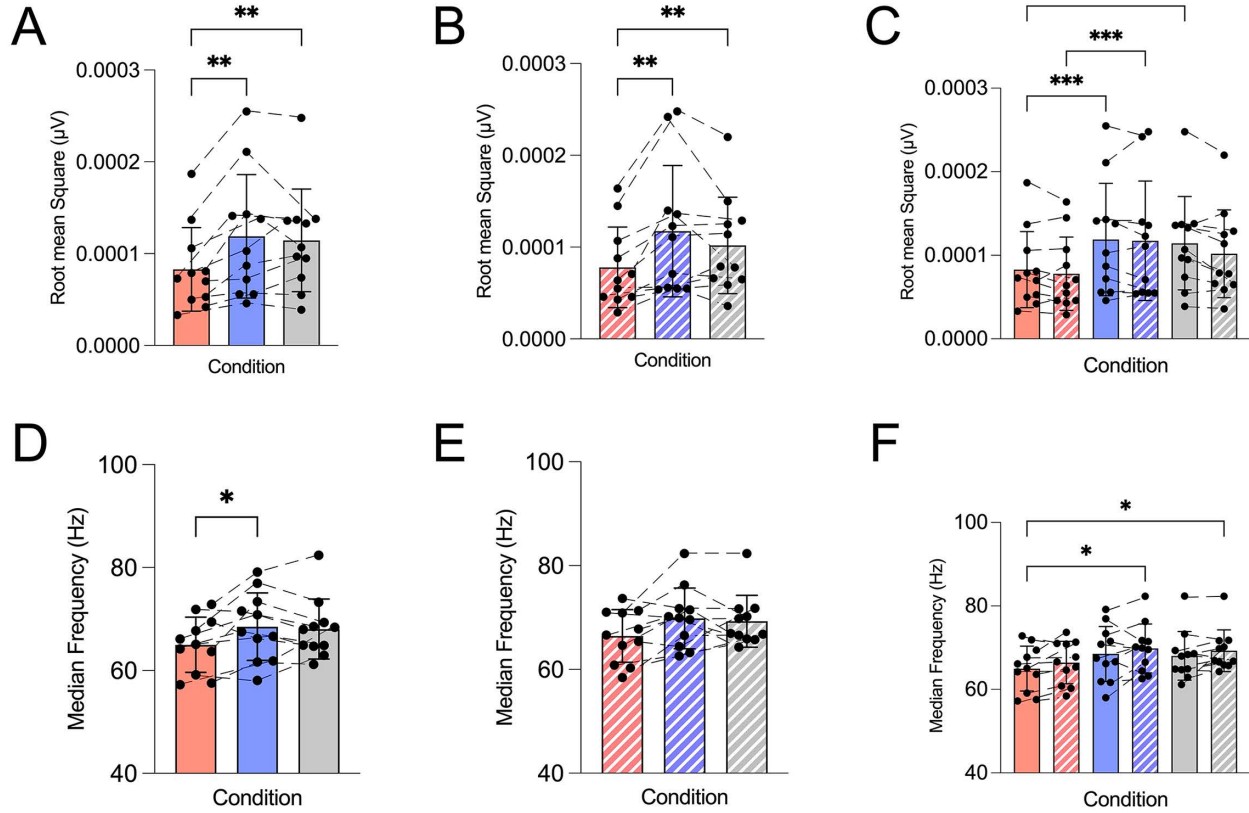

**Fig 3. Effect of time-of-day and caffeine ingestion on neuromuscular activation.** Morning caffeine (AM$_{CAFF}$) increased EMG amplitude compared with morning placebo (AM$_{PLAC}$), with no clear changes in median frequency, indicating enhanced neural drive without alterations in frequency-domain fatigue characteristics. Data are presented as mean±SD with individual values that represent single participants plotted. Trials were conducted in the morning (08:00) with placebo (AM$_{PLAC}$), with caffeine ingestion (6 mg.kg$^{-1}$; AM$_{CAFF}$), and in the evening (17:00; PM$_{PLAC}$). Red bars represent AM$_{PLAC}$, blue bars AM$_{CAFF}$, and grey bars PM$_{PLAC}$. Solid bars indicate pre-fatigue MVIC, and hatched bars indicate post-fatigue MVIC. **(A-C)** Root mean square EMG amplitude measured in µV during pre- and post-fatigue MVIC. **(D-F)** Median frequency measured in Hz during pre- and post-fatigue MVIC. Statistical analysis was performed using one-way and two-way repeated-measures ANOVA with Tukey's post hoc tests. Significance is indicated as *$p < 0.05$, **$p < 0.01$, ***$p < 0.001$, ****$p < 0.0001$.

### Rate of perceived exertion

RPE (Fig 4A) showed a main effect of condition ($p < 0.01$), and post-hoc analysis revealed that ratings were greater in AM$_{PLAC}$ (7.92±0.49) compared with both AM$_{CAFF}$ (6.23±0.93) and PM$_{PLAC}$ (6.31±0.75), with values ~20% lower in both trials (both $p < 0.01$). No differences ($p > 0.05$) were measured between AM$_{CAFF}$ and PM$_{PLAC}$.

### Core temperature

Core temperature (Fig 4B) showed a main effect of condition ($p < 0.05$), and post-hoc analysis indicated that PM$_{PLAC}$ (36.10±0.64 °C) was almost 3% higher than both AM$_{PLAC}$ (35.32±0.61 °C) and AM$_{CAFF}$ (35.52±0.66 °C), both $p < 0.05$. There were no differences ($p > 0.05$) detected between the two morning conditions.

### Discussion

The present study demonstrates that 6 mg·kg$^{-1}$ of caffeine ingestion 45 minutes before exercise attenuates diurnal decline in morning neuromuscular performance in healthy young women. Morning caffeine intake (AM$_{CAFF}$) significantly

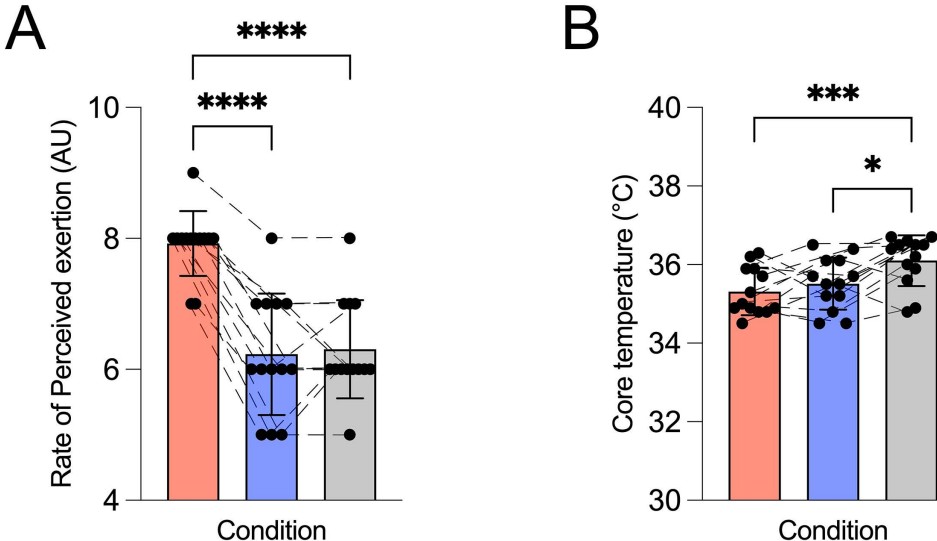

**Fig 4. Effect of time-of-day and caffeine ingestion on perception of effort and core temperature.** Morning caffeine ($AM_{CAFF}$) reduced perceived exertion compared with morning placebo ($AM_{PLAC}$), with no meaningful differences in core temperature between conditions. Data are presented as mean±SD with individual values that represent single participants plotted. Trials were conducted in the morning (08:00) with placebo ($AM_{PLAC}$), with caffeine ingestion (6 mg.kg⁻¹; $AM_{CAFF}$), and in the evening (17:00; $PM_{PLAC}$). Red bars represent $AM_{PLAC}$, blue bars $AM_{CAFF}$, and grey bars $PM_{PLAC}$. **(A)** Rating of perceived effort (scale 1-10) was taken during each trial, and **(B)** tympanic temperature was measured in degrees Celsius. Statistical analysis was performed using one-way repeated-measures ANOVA with Tukey's post hoc testing. Significance is indicated as *$p < 0.05$, **$p < 0.01$, ***$p < 0.001$, ****$p < 0.0001$.

increased both peak torque by 33–45%, and time to exhaustion by 43% compared with morning placebo ($AM_{PLAC}$), elevating both outcomes to levels observed in the evening trial ($PM_{PLAC}$) (Fig 2). Importantly, given the study design, these findings reflect between-condition differences rather than within-session pre-post effects of caffeine ingestion. This improvement was accompanied by higher EMG amplitude (Fig 3), and lower ratings of perceived exertion (Fig 4), while median frequency of EMG and tympanic temperature remained the same (Figs 3 and 4). The intermittent nature of the fatigue protocol may permit partial recovery between contractions and introduce pacing strategies, which could contribute to the preservation of post-fatigue maximal voluntary isometric contraction (MVIC) values, as previously described in intermittent isometric fatigue models [51]. Our data are consistent with the well-established evidence [1–3,7–9] of diurnal performance differences in physical performance variation in neuromuscular performance, with peak torque and time to exhaustion consistently lower in the morning relative to the evening placebo conditions. The magnitude of these improvements is comparable to, and in some cases greater than, those reported in previous studies examining caffeine's effects on strength and endurance performance conducted later in the day [27,28], suggesting that caffeine may be particularly effective in mitigating circadian-related performance decrements when exercised in the morning. However, the magnitude observed here (~20–30%) is relatively large compared to typical reports in the literature, where improvements in neuromuscular performance following caffeine ingestion are more commonly in the range of ~5–10% [41,42], suggesting that additional factors such as time-of-day or within subject variability may have contributed. Collectively, our data indicate that performance in the caffeine condition was comparable to evening levels in neuromuscular performance, primarily via central mechanisms rather than peripheral or thermoregulatory processes.

Previous investigations of diurnal differences in performance have attempted to attribute daily fluctuations in performance to core temperature [2,16–19], metabolic activity [54], and neural function [55,56]. We investigated the effect of ergogenic aid consumption and associated effects of core temperature and neuromuscular function. Importantly, our data

show that in females caffeine ingestion can elevate morning performance to evening levels, in agreement with previous work demonstrating that caffeine mitigates circadian-related decrements in strength and power of male athletes [31,32]. Meta-analyses further highlight consistent ergogenic effects across both strength and endurance tasks [27,28,57], supporting the robustness of the current findings. For example, the present study adds new evidence that this effect also occurs in females, suggesting the mechanism is centrally mediated and independent of temperature.

Not all studies have observed uniform effects. Some evidence suggests that caffeine exerts greater benefits in tasks involving larger muscle groups [33], or dynamic contractions [58]. For example, Mora-Rodríguez et al, [31] reported improvements in dynamic but not isometric contractions at 3 mg·kg⁻¹, whereas benefits at 6 mg·kg⁻¹ were evident in maximal squats, but not bench press [32]. In contrast to studies reporting limited or task-specific effects of caffeine on isometric performance [31,32], the present study demonstrates clear improvements in isometric knee extensor torque and fatigue tolerance, likely reflecting the use of a large lower-limb muscle group, a higher caffeine dose (6 mg·kg⁻¹), and the inclusion of sustained maximal contractions. This adds to the literature by showing that caffeine can also enhance isometric performance under fatigue when sufficient dose and muscle mass engaged.

The absence of a direct association between core temperature and performance provides mechanistic insight. As expected, evening trials were accompanied by elevated temperature, consistent with circadian regulation [11]. Yet, caffeine improved torque and time to exhaustion in the morning without increasing temperature, suggesting that thermoregulatory changes were not responsible for the ergogenic effect in our data. This further aligns with previous work showing caffeine can improve neuromuscular performance independent of body temperature [31,32], and supports a centrally mediated mechanism now demonstrated for the first time in females. Future studies incorporating larger sample sizes may benefit from examining associations between diurnal changes in physiological variables, e.g., temperature, EMG characteristics, and performance outcomes to further elucidate the mechanisms underlying the ergogenic effects of caffeine.

Instead, our EMG findings suggest enhanced central neural drive. RMS amplitude, an index of motor unit recruitment and activation, was higher with caffeine ingestion, while MDF, a spectral index related to motor unit firing behaviour and fatigue, remained unchanged. This pattern indicates more sustained motor unit activation rather than altered peripheral conduction, in line with earlier reports showing caffeine-induced increases in EMG amplitude without significant changes in spectral fatigue markers [22,26,49], and extending these observations to a circadian context in females. These results are consistent with the established central actions of caffeine as an adenosine receptor antagonist, which increases cortical excitability, augments neurotransmitter release, and reduces perceived exertion [59,60]. The significant reduction in RPE observed here further supports this central mechanism, consistent with reports that caffeine lowers perception of effort and enhances tolerance to fatigue [9,10,49,61].

A critical contribution of this study is its focus on females, a population historically under-represented in caffeine research [38]. Sex hormones influence caffeine metabolism via modulation of CYP1A2 activity, with menstrual cycle timing and contraceptive use known to affect clearance rates and plasma exposure [36,37]. Hormonal fluctuations also influence thermoregulation, substrate metabolism, and neuromuscular efficiency. In the present study, caffeine ingestion improved torque, time to exhaustion, and EMG amplitude in the morning, indicating enhanced neuromuscular performance under these conditions. Whilst these findings are consistent with a centrally mediated ergogenic effect, the extent to which hormonal factors contributed to inter-individual variability cannot be determined from the current data.

Despite these complexities, most caffeine research has been male-centric or insufficiently controlled for female-specific physiology. By recruiting healthy young females and standardising testing based on menstrual cycle timing, the present study addresses this gap and provides evidence that caffeine improves morning neuromuscular performance in this group. However, menstrual cycle timing was determined via self-report, and no objective verification, e.g., hormonal assays or ovulation testing, was performed. Therefore, true cycle phase cannot be confirmed and may have contributed to inter-individual variability [62]. Future studies should incorporate objective hormonal verification to better characterise the interaction between menstrual cycle dynamics and caffeine responses, and expand on this work to older or

post-menopausal women, where reduced oestrogen and progesterone levels may alter the pharmacokinetics and efficacy of caffeine [63,64].

Several limitations warrant consideration. First, while participants refrained from caffeine for 48 hours prior to testing, habitual caffeine intake was not fully quantified, leaving potential residual effects. Second, the fixed order of testing represents a limitation, as order effects cannot be fully excluded; however, this approach was necessary to preserve circadian timing and minimise potential caffeine carryover between conditions. The study did not include a within-session pre-ingestion MVIC assessment in the caffeine condition, which may limit the ability to evaluate acute within-session changes. However, the crossover design allows for controlled between-condition comparisons under matched circadian conditions. Chronotype and sleep were not rigorously controlled, each of which can modulate circadian physiology and caffeine metabolism [21,65]. Third, the single-blind design carries risk of expectancy bias, as the physiological side effects of caffeine may have reduced masking integrity [35]. Blinding efficacy was not formally assessed, e.g., condition guessing, which may have influenced subjective outcomes such as ratings of perceived exertion. Plasma caffeine concentrations were also not measured, preventing direct verification of absorption kinetics. The relatively small sample size should be considered when interpreting the findings. Although the repeated-measures crossover design enhances statistical power by reducing inter-individual variability, the modest sample size may limit generalisability and the detection of smaller effects. Finally, EMG provides indirect insight into neuromuscular activation, but more definitive approaches such as interpolated twitch techniques or transcranial magnetic stimulation would better disentangle central versus peripheral contributions [66].

Despite these limitations, the findings offer important contributions to both theory and practice. Theoretically, these data support the notion that the ergogenic potential of caffeine in the morning operates primarily through central rather than thermal or peripheral pathways. Practically, they suggest two strategies to mitigate diurnal performance decrements: i) scheduling training or competition in the evening to align with circadian peaks, or ii) ingesting caffeine (6 mg·kg$^{-1}$) 45–60 minutes before morning exercise. Given inter-individual variability in caffeine sensitivity, practitioners should note that the dose used in the present study (6 mg·kg$^{-1}$) is relatively high and may not be suitable for all individuals. Accordingly, caffeine dosing should be tailored (3–6 mg·kg$^{-1}$) based on individual tolerance and habitual intake, with careful monitoring for adverse effects. Finally, although the present sample consisted of young adults, understanding how caffeine interacts with circadian neuromuscular regulation in females provides a mechanistic foundation that may inform future work in ageing populations, where circadian amplitude, neuromuscular function, and caffeine metabolism are altered.

From a broader perspective, future work may benefit from modelling the temporal dynamics of neuromuscular performance in relation to circadian rhythms and ergogenic interventions such as caffeine, similar to approaches applied in domains such as vigilance [67]. Such frameworks may help to better characterise the interaction between biological rhythms and performance outcomes over time.

## Conclusion

In summary, acute caffeine ingestion (6 mg·kg$^{-1}$) taken 45 minutes before exercise results in morning neuromuscular performance that is comparable to evening levels in young females. Improvements in torque, time to exhaustion, EMG amplitude, and perceived exertion occurred without changes in EMG frequency characteristics or core temperature, indicating a central rather than peripheral or thermoregulatory mechanism. These findings extend existing literature by demonstrating for the first time that caffeine effectively offsets circadian-related performance decrements in females, a group underrepresented in scientific research.

## Author contributions

**Conceptualization:** Stuart J. Hesketh.

**Data curation:** Stuart J. Hesketh.

**Investigation:** Akshay Singh.

**Methodology:** Stuart J. Hesketh.

**Project administration:** Akshay Singh.

**Supervision:** Stuart J. Hesketh.

**Visualization:** Stuart J. Hesketh.

**Writing – original draft:** Akshay Singh.

**Writing – review & editing:** Stuart J. Hesketh.

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
