## [Decision Letter · Decision Letter 0]

14 Jan 2026

PONE-D-25-56298Caffeine ingestion restores morning neuromuscular performance to evening levels in healthy females: A randomised crossover studyPLOS One

Dear Dr. Hesketh

Thank you for submitting your manuscript to PLOS ONE. After careful consideration, we feel that it has merit but does not fully meet PLOS ONE’s publication criteria as it currently stands. Therefore, we invite you to submit a revised version of the manuscript that addresses the points raised during the review process. Please submit your revised manuscript by Feb 28 2026 11:59PM. If you will need more time than this to complete your revisions, please reply to this message or contact the journal office at plosone@plos.org. Please include the following items when submitting your revised manuscript:

We look forward to receiving your revised manuscript.

Kind regards,

Alberto Souza Sá Filho, Ph.D

Academic Editor

PLOS One

Journal Requirements:

Additional Editor Comments (if provided):

Dear Authors, congratulations on your work! We will continue with the paper's development process. Therefore, please heed the feedback from our esteemed reviewers. If you can, you will improve the quality of your work. I want to make it clear that the possibility of new rounds exists, therefore, do your best to close any gaps. Thanks!

Reviewers' comments:

Reviewer's Responses to Questions

**Comments to the Author**

1. Is the manuscript technically sound, and do the data support the conclusions?

Reviewer #1: No

Reviewer #2: Yes

2. Has the statistical analysis been performed appropriately and rigorously?

Reviewer #1: Yes

Reviewer #2: Yes

3. Have the authors made all data underlying the findings in their manuscript fully available?

Reviewer #1: Yes

Reviewer #2: Yes

4. Is the manuscript presented in an intelligible fashion and written in standard English?

Reviewer #1: Yes

Reviewer #2: Yes

5. Review Comments to the Author

Reviewer #1: This study is relevant to the scientific community specifically in the area of exercise physiology and health.

Therefore, some considerations regarding the study methodology were made:

1) The distribution of the sample into groups did not include an isolated control group. Since each participant was controlled by themselves, at what point were the data relating to the control phase of the study collected? Include this data in the statistics for comparison with the other groups.

2) It was reported that there were 3 evaluation sessions. Was there a data collection session solely for baseline measurements? If this collection occurred in the same performance session, what was the interval between the end of the baseline measurement collection and the beginning of the performance data collection?

3) Provide a reference regarding the collection of tympanic temperature. Why was this measurement chosen?

4) Regarding electromyography, how many electrodes were used? Why was the vastus medialis muscle chosen? Why was only this muscle chosen and not others of the quadriceps femoris?

5) What were the inclusion and exclusion criteria for the sample?

6) How was randomization performed?

7) What was the fitness level of the participants? Had the sample previously participated in a strength training program?

8) Why wasn't the sample's BMI calculated for comparison purposes?

9) Include the assessment of neuromuscular function under fatigue conditions in the last paragraph of the introduction.

10) The graphs should be self-explanatory. I suggest creating a legend for each group and for each symbol used.

11) What were the results of the effect sizes of the research?

14) In the Discussion, you should compare the results found with studies already conducted on the topic and not just describe your results. This is the part of the manuscript where you should cite other authors of related studies for comparison purposes.

Reviewer #2: Thank you for the opportunity to review this interesting and timely manuscript examining whether acute caffeine ingestion can restore morning neuromuscular performance to evening levels in healthy young females. The topic is relevant to exercise physiology, chronobiology, sport science, and women’s health research, and the manuscript is generally clear, well written, and logically structured. Importantly, it addresses a recognised gap in the literature: the underrepresentation of female participants in caffeine research and the complexity introduced by menstrual cycle-related hormonal fluctuations.

The study provides valuable data on the interaction between caffeine, time-of-day, and neuromuscular function, offering potentially meaningful practical implications for morning performance optimisation. The experimental design is conceptually sound, the measurements are appropriate for the research question, and the authors articulate their findings coherently. The manuscript therefore has clear potential for publication; however, several methodological and reporting issues reduce the current level of rigour and require clarification or expansion before the manuscript can be considered suitable for PLOS Aging and Health.

Below I outline my comments in detail, separating general concerns from specific points requiring revision.

The study demonstrates adequate originality for the journal, particularly in its focus on young women—a population that remains understudied in caffeine-related performance research. The chronobiological framing adds conceptual value and the results complement existing knowledge regarding diurnal variations in strength, endurance, and neuromuscular activation.

Nevertheless, certain aspects limit the broader impact expected for a publication in PLOS Aging and Health: chiefly the small sample size, limited generalisability beyond young adults, and the narrow task specificity (isometric knee extensor MVC and endurance tests). Strengthening the rationale and contextualisation in terms of lifespan health and sex-specific physiology would enhance the manuscript’s relevance for readers across the ageing and health sciences spectrum.

Importantly, although the study meets ethical standards and uses recognised methodologies, several key elements require more detailed reporting to satisfy the journal’s requirement for high methodological rigour and substantial evidence supporting the conclusions.

I recommend moderate revisions.

The manuscript states that 13 healthy young females participated and were classified as light caffeine consumers. While this provides a general notion of the participant pool, essential details regarding the physical activity level and training status are missing. These factors significantly affect neuromuscular performance, caffeine responsiveness, fatigue tolerance, and task familiarity. Without them, generalisability is unclear and between-participant variability cannot be fully contextualised.

Recommended revision:

Please provide explicit information on the participants’ habitual physical activity, including estimated weekly volume, type of exercise (aerobic, resistance, recreational sport), and whether any had experience with isometric or strength testing. This information should also be incorporated into the inclusion criteria.

Additionally, please list inclusion and exclusion criteria explicitly. At present, these must be inferred (health status, menstrual phase, caffeine intake, etc.), but explicit statements would improve transparency and reproducibility.

The manuscript notes that testing occurred in the mid-follicular phase (~days 10–15), which is appropriate given the need to minimise hormonal variation. However, it is not stated how this phase was determined. Self-report, cycle diary, ovulation predictors, or hormonal assays each have different levels of reliability. Moreover, it is not indicated whether hormonal contraceptive users were excluded, even though contraceptives may substantially affect caffeine metabolism (via modulation of CYP1A2), thermoregulation, substrate utilisation, and neuromuscular efficiency.

Given the known importance of hormonal dynamics in caffeine studies involving women, these omissions weaken the methodological transparency and must be addressed.

Recommended revision:

• Specify explicitly how menstrual cycle phase was determined (e.g., self-reported cycle day, tracking apps, hormonal confirmation).

• State whether hormonal contraceptive use was an exclusion criterion. If not, please clarify whether any participants used contraceptives and include this as a limitation.

This is essential for reproducing the study and interpreting intra- and inter-participant variability.

The manuscript instructs participants to abstain from caffeine, alcohol, smoking, and strenuous exercise for 48 hours prior to each trial. However, the experimental protocol involves three sessions on consecutive days, making it impossible for participants to meet this requirement before each session unless the authors intended abstinence to begin 48 hours before the first session and continue throughout.

Furthermore, because caffeine was administered in only one condition, and sessions occurred within a short time frame, it is crucial to clarify whether:

• the caffeine session was always placed last, or

• conditions were randomised, in which case absence of a washout period becomes a significant methodological concern.

This requires explicit correction.

Recommended revision:

• Clarify whether abstinence was continuous throughout the three days rather than reset before each session.

• Specify the order of the three conditions (Morning Placebo, Morning Caffeine, Evening Placebo).

• If randomised, acknowledge that a washout period longer than 24 hours may be required to eliminate residual caffeine effects, especially given inter-individual differences in CYP1A2 activity.

• Discuss this explicitly in the Limitations section.

This is important because caffeine’s half-life, combined with individual metabolic variability, may lead to carry-over effects.

The use of a single-blind design is stated, but the manuscript does not reflect on the implications. Given that caffeine often produces noticeable physiological sensations (increased alertness, heart rate changes, etc.), the blinding integrity may have been compromised.

Furthermore, no assessment of blinding efficacy (e.g., asking participants to guess their condition) is reported. This weakens interpretability, particularly for subjective outcomes such as perceived exertion.

Recommended revision:

Please discuss the limitations of the single-blind design more explicitly and consider adding a note in the Limitations section regarding potential expectancy bias. While not fatal to the study, acknowledging this improves transparency.

The choice of 6 mg/kg is common in ergogenic research and the ingestion timing (45 minutes pre-test) aligns with established pharmacokinetics. However, providing a brief justification in the Methods section, referencing plasma peak timing, would strengthen methodological clarity.

Additionally, in the Practical Applications or Discussion section, please briefly acknowledge that this dose is relatively high and may not be suitable for all individuals. This will help meet the journal’s requirement for clear utility for the broader community, especially given differences in caffeine sensitivity among women.

The use of isometric knee extensor tasks is well established, but the manuscript does not explicitly justify this choice. Although the rationale can be inferred (large muscle group, reliable EMG signal, widely used in chronobiology and strength research), a concise explanation would improve clarity.

Recommended revision:

Add 1–2 sentences describing why knee extensors were selected, emphasising reliability, comparability to prior studies, and suitability for assessing neuromuscular responses to circadian and caffeine-related factors.

The manuscript does not provide a sample size justification or a power analysis. Although crossover designs reduce inter-individual variability, a formal justification is expected for publication in PLOS journals.

Recommended revision:

Include whether an a priori or post hoc power estimation was performed. If not feasible, please provide a short statement acknowledging this and discussing the implications.

This study addresses a meaningful and underexplored topic, contributes valuable data to the field, and offers potential practical relevance. With revisions addressing methodological clarity, enhanced description of participant characteristics, improved explanation of menstrual cycle control, and more cautious interpretation of mechanisms, the manuscript will meet the standards required for publication in PLOS Aging and Health.

I encourage the authors to revise the manuscript according to the points listed above. The core of the study is strong, and I believe it has good potential for publication after these improvements.

6. PLOS authors have the option to publish the peer review history of their article (what does this mean?). If published, this will include your full peer review and any attached files.

Reviewer #1: **Yes:** Elciana de Paiva Lima Vieira

Reviewer #2: **Yes:** Ana Isabel Vieira

---

## [Author Response · Author response to Decision Letter 1]

11 Feb 2026

manuscript and for their constructive and insightful comments. We have revised the manuscript accordingly and believe that these changes have strengthened the clarity, methodological rigour, and overall quality of the work. A detailed, point-by-point response to each comment is provided below. Reviewer comments are reproduced in black, and our responses are provided in bold red text beneath each comment. All changes made to the manuscript are reflected in the revised version.

Reviewer #1: This study is relevant to the scientific community specifically in the area of exercise physiology and health.

We thank Reviewer #1 for their constructive feedback and for recognising the relevance of the study to exercise physiology and health research. All comments are addressed individually below.

Therefore, some considerations regarding the study methodology were made:

1) The distribution of the sample into groups did not include an isolated control group. Since each participant was controlled by themselves, at what point were the data relating to the control phase of the study collected? Include this data in the statistics for comparison with the other groups.

We appreciate this comment and have clarified the within-subject control structure of the study. As a repeated-measures crossover design was employed, an isolated control group was not required. The morning placebo (AMPLAC) and evening placebo (PMPLAC) conditions served as control phases, enabling direct comparison of time-of-day effects and caffeine ingestion within the same participants. Data from all placebo and caffeine conditions were included in the statistical analyses and are presented in Table 1 and Figures 2–4. New text has also been added to the ‘Study design and Supplementation section in the methods to help with clarity.

2) It was reported that there were 3 evaluation sessions. Was there a data collection session solely for baseline measurements? If this collection occurred in the same performance session, what was the interval between the end of the baseline measurement collection and the beginning of the performance data collection?

Baseline measurements were collected at the start of each experimental session and not during a separate visit. These measurements were immediately followed by a standardised warm-up, after which performance testing commenced. Wording has also been changed in the experimental protocol section of the methods to ensure clarity of timing between baseline collection and beginning of warm-up and experimental procedures.

3) Provide a reference regarding the collection of tympanic temperature. Why was this measurement chosen?

Tympanic temperature was selected as a non-invasive and practical surrogate of core temperature, suitable for repeated measurements across multiple experimental sessions. Tympanic thermometry is widely used in exercise and circadian research and has been shown to reflect changes in core body temperature under resting and exercise conditions. Appropriate reference (PMID: 12427049) has now been added along with new text in the experimental protocol section of the methods to support the validity and rationale for this methodological choice.

4) Regarding electromyography, how many electrodes were used? Why was the vastus medialis muscle chosen? Why was only this muscle chosen and not others of the quadriceps femoris?

Surface electromyography was recorded using bipolar electrodes placed over a single muscle (vastus medialis) in accordance with SENIAM guidelines. The vastus medialis was selected due to its major contribution to knee extensor torque during isometric contractions, its superficial anatomy allowing reliable surface EMG recordings, and its frequent use in studies examining neuromuscular activation and fatigue of the knee extensors. Recording EMG from a single representative quadriceps muscle is a common and accepted approach when the primary aim is to assess relative changes in neuromuscular activation across conditions. Limiting recordings to one muscle also reduces cross-talk, improves signal quality, and enhances repeatability across sessions. Appropriate justification and references have now been added to the Neuromuscular testing section in the Methods.

5) What were the inclusion and exclusion criteria for the sample?

We thank the reviewer for highlighting the need for explicit reporting. Inclusion and exclusion criteria have now been clearly specified in the Subjects section of the methods, including health status, caffeine consumption, menstrual cycle considerations, and exercise- and medication-related restrictions.

6) How was randomization performed?

We thank the reviewer for highlighting the importance of clearly reporting the testing sequence, which has helped improve the methodological transparency of the manuscript. The order of experimental conditions was not randomised; instead, a fixed testing sequence was deliberately employed to accommodate the circadian nature of the study design and to minimise potential caffeine carryover effects between sessions. Specifically, the evening placebo condition was completed separately from the morning trials, and the morning caffeine condition was performed after the morning placebo condition to avoid residual caffeine influencing placebo measurements. We believe this fixed sequence was an appropriate and conservative design choice for addressing the study aims. The testing order has now been explicitly clarified in the study design and supplementation section of the Methods, and any wording implying randomisation has been removed including the title. This aspect of the design has also been acknowledged in the Limitations section.

7) What was the fitness level of the participants? Had the sample previously participated in a strength training program?

We thank the reviewer for highlighting the importance of describing participant fitness and training background. Participants were recreationally active but not engaged in structured resistance or strength training programmes, and none had prior experience with isometric strength testing. This information has now been explicitly added to the Subjects section of the Methods.

8) Why wasn't the sample's BMI calculated for comparison purposes?

We thank the reviewer for this observation. Body mass was recorded and peak torque was normalised to body mass, which we considered more physiologically relevant for interpreting neuromuscular performance outcomes than BMI. BMI does not account for body composition and is less informative for strength-related measures in healthy, non-obese participants. As such, BMI was not included as a primary descriptive or comparative variable. This rationale has now been clarified in the statistical analysis section of the Methods.

9) Include the assessment of neuromuscular function under fatigue conditions in the last paragraph of the introduction.

We thank the reviewer for this helpful suggestion. The final paragraph of the Introduction has now been revised to explicitly include the assessment of neuromuscular function under fatigued conditions, in line with the experimental design and outcomes measured in the study.

10) The graphs should be self-explanatory. I suggest creating a legend for each group and for each symbol used.

We thank the reviewer for this suggestion. All figures were designed to be stand-alone and include condition labels, colour coding, and symbol definitions within the legends. To further improve clarity, the figure legends have been slightly expanded to explicitly describe group coding and the meaning of individual data points and bar shading. No changes to the data or graphical presentation were required.

11) What were the results of the effect sizes of the research?

We thank the reviewer for this comment. Effect sizes (η² and partial η²) were calculated for all main and interaction effects and are reported in Table 1 alongside inferential statistics. These values indicate moderate to large effects across the primary neuromuscular outcomes, supporting the practical relevance of the findings. To improve clarity, this has now been briefly highlighted in the Results section were Table 1 is introduced.

14) In the Discussion, you should compare the results found with studies already conducted on the topic and not just describe your results. This is the part of the manuscript where you should cite other authors of related studies for comparison purposes.

We thank the reviewer for this comment. The Discussion has been revised to more explicitly compare the present findings with previously published studies, highlighting areas of agreement and divergence with existing literature, particularly with respect to task specificity, effect magnitude, and neuromuscular mechanisms.

Reviewer #2: Thank you for the opportunity to review this interesting and timely manuscript examining whether acute caffeine ingestion can restore morning neuromuscular performance to evening levels in healthy young females. The topic is relevant to exercise physiology, chronobiology, sport science, and women’s health research, and the manuscript is generally clear, well written, and logically structured. Importantly, it addresses a recognised gap in the literature: the underrepresentation of female participants in caffeine research and the complexity introduced by menstrual cycle-related hormonal fluctuations.

The study provides valuable data on the interaction between caffeine, time-of-day, and neuromuscular function, offering potentially meaningful practical implications for morning performance optimisation. The experimental design is conceptually sound, the measurements are appropriate for the research question, and the authors articulate their findings coherently. The manuscript therefore has clear potential for publication; however, several methodological and reporting issues reduce the current level of rigour and require clarification or expansion before the manuscript can be considered suitable for PLOS Aging and Health.

Below I outline my comments in detail, separating general concerns from specific points requiring revision. The study demonstrates adequate originality for the journal, particularly in its focus on young women—a population that remains understudied in caffeine-related performance research. The chronobiological framing adds conceptual value and the results complement existing knowledge regarding diurnal variations in strength, endurance, and neuromuscular activation.

Nevertheless, certain aspects limit the broader impact expected for a publication in PLOS Aging and Health: chiefly the small sample size, limited generalisability beyond young adults, and the narrow task specificity (isometric knee extensor MVC and endurance tests). Strengthening the rationale and contextualisation in terms of lifespan health and sex-specific physiology would enhance the manuscript’s relevance for readers across the ageing and health sciences spectrum.

Importantly, although the study meets ethical standards and uses recognised methodologies, several key elements require more detailed reporting to satisfy the journal’s requirement for high methodological rigour and substantial evidence supporting the conclusions.

We thank Reviewer #2 for their thoughtful, detailed, and constructive evaluation of our manuscript. We appreciate their positive assessment of the study’s relevance, conceptual framing, and contribution to addressing the underrepresentation of females in caffeine research. We also acknowledge the reviewer’s important points regarding methodological transparency, reporting clarity, and broader contextualisation within ageing and health sciences. In response, we have revised the manuscript to provide additional methodological detail, clearer justification of design choices, and strengthened discussion of the findings in relation to sex-specific physiology and lifespan health considerations. Each comment is addressed individually below.

I recommend moderate revisions.

The manuscript states that 13 healthy young females participated and were classified as light caffeine consumers. While this provides a general notion of the participant pool, essential details regarding the physical activity level and training status are missing. These factors significantly affect neuromuscular performance, caffeine responsiveness, fatigue tolerance, and task familiarity. Without them, generalisability is unclear and between-participant variability cannot be fully contextualised.

Recommended revision:

Please provide explicit information on the participants’ habitual physical activity, including estimated weekly volume, type of exercise (aerobic, resistance, recreational sport), and whether any had experience with isometric or strength testing. This information should also be incorporated into the inclusion criteria.

Additionally, please list inclusion and exclusion criteria explicitly. At present, these must be inferred (health status, menstrual phase, caffeine intake, etc.), but explicit statements would improve transparency and reproducibility.

We thank the reviewer for highlighting the importance of clearly characterising participants’ physical activity and training background. In response, we have expanded the Subjects section of the methods to explicitly describe habitual physical activity level and prior resistance or isometric exercise experience. Participants were recreationally active but not engaged in structured strength or resistance training programmes, and none had prior experience with isometric knee extensor testing beyond familiarisation. In addition, inclusion and exclusion criteria have now been explicitly listed in the Subjects section, including health status, caffeine consumption, menstrual cycle considerations, physical activity level, and medication or supplement use. These revisions improve transparency, reproducibility, and contextualisation of inter-individual variability.

The manuscript notes that testing occurred in the mid-follicular phase (~days 10–15), which is appropriate given the need to minimise hormonal variation. However, it is not stated how this phase was determined. Self-report, cycle diary, ovulation predictors, or hormonal assays each have different levels of reliability. Moreover, it is not indicated whether hormonal contraceptive users were excluded, even though contraceptives may substantially affect caffeine metabolism (via modulation of CYP1A2), thermoregulation, substrate utilisation, and neuromuscular efficiency.

Given the known importance of hormonal dynamics in caffeine studies involving women, these omissions weaken the methodological transparency and must be addressed.

Recommended revision:

• Specify explicitly how menstrual cycle phase was determined (e.g., self-reported cycle day, tracking apps, hormonal confirmation).

• State whether hormonal contraceptive use was an exclusion criterion. If not, please clarify whether any participants used contraceptives and include this as a limitation.

This is essential for reproducing the study and interpreting intra- and inter-participant variability.

We thank the reviewer for highlighting the importance of clearly reporting menstrual cycle determination and contraceptive status. Menstrual cycle phase was determined using self-reported cycle length and day of cycle, confirmed verbally at each visit, with all testing scheduled during the mid-follicular phase (~days 10–15). This approach has been explicitly stated in the Subjects section of the Methods. Hormonal contraceptive use was an exclusion criterion, and no participants were using oral or other hormonal contraceptives. This has now been clearly reported in the exclusion criteria. These additions improve methodological transparency and support reproducibility and interpretation of inter-individual variability.

The manuscript instructs participants to abstain from caffeine, alcohol, smoking, and strenuous exercise for 48 hours prior to each trial. However, the experimental protocol involves three sessions on consecutive days, making it impossible for participants to meet this requirement before each session unless the authors intended abstinence to begin 48 hours before the first session and continue throughout.

Furthermore, because

---

## [Decision Letter · Decision Letter 1]

6 Apr 2026

PONE-D-25-56298R1Caffeine ingestion restores morning neuromuscular performance to evening levels in healthy femalesPLOS One

Dear Dr. Hesketh,

Thank you for submitting your manuscript to PLOS ONE. I was re-assigned as academic editor for this submission, congratulations for your work. I invited another expert in neuromuscular function and menstrual cycle, which provided pertinent comments and evidenced important concerns that need to be addressed to make the work suitable for publication. After careful consideration, we feel that it has merit but does not fully meet PLOS ONE’s publication criteria as it currently stands. Therefore, we invite you to submit a revised version of the manuscript that addresses the points raised during the review process.

We look forward to receiving your revised manuscript.

Kind regards,

Giorgio Varesco, PhD

Academic Editor

PLOS One

Journal Requirements:

Additional Editor Comments :

Please, also consider these additional comments to improve the quality of your figures:

I suggest adjusting Figure 1 to reflect the right order and mention Day1, day2, day3.

Please reduce the bar line thickness in your figures, add lines connecting dots to see individual trajectories. Please rescale figures 3 d,e and f; and figure 4 a, b.

Reviewers' comments:

Reviewer's Responses to Questions

**Comments to the Author**

1. If the authors have adequately addressed your comments raised in a previous round of review and you feel that this manuscript is now acceptable for publication, you may indicate that here to bypass the “Comments to the Author” section, enter your conflict of interest statement in the “Confidential to Editor” section, and submit your "Accept" recommendation.

Reviewer #2: All comments have been addressed

Reviewer #3: (No Response)

2. Is the manuscript technically sound, and do the data support the conclusions?

Reviewer #2: Yes

Reviewer #3: Partly

3. Has the statistical analysis been performed appropriately and rigorously?

Reviewer #2: Yes

Reviewer #3: Yes

4. Have the authors made all data underlying the findings in their manuscript fully available?

Reviewer #2: Yes

Reviewer #3: Yes

5. Is the manuscript presented in an intelligible fashion and written in standard English?

Reviewer #2: Yes

Reviewer #3: Yes

6. Review Comments to the Author

Reviewer #2: The revised manuscript shows a clear and consistent effort by the authors to address the reviewers’ comments. The changes made have substantially improved the quality of the paper, particularly in terms of methodological clarity and transparency in reporting procedures and study limitations.

Although some limitations remain inherent to the study design, these are appropriately acknowledged and discussed by the authors, and do not compromise the overall coherence or scientific contribution of the work.

In this context, I consider that the manuscript meets the requirements for publication and recommend its acceptance.

Reviewer #3: The present manuscript investigated the effects of ingesting 6 mg·kg⁻¹ of caffeine 45 minutes prior to a morning exercise session in 13 active female participants. The aim was to determine whether caffeine could increase maximal voluntary isometric contraction (MVIC) and fatigue resistance in the morning to levels comparable to those observed in the evening. It reported that caffeine ingestion resulted in improvements in MVIC, time to exhaustion, EMG amplitude, and perceived exertion, without changes in EMG frequency characteristics or core temperature compared to placebo, with similar results to evening levels. I appreciate that this study evaluates females when the literature is historically biased on males-only studies, and I read it with interest.

However, there are important concerns that must be addressed:

1. How do you explain that maximal torque after the task was practically identical to “pre-fatigue” when by design it should have drop below 50%? Pacing could be involved. To clarify you could consider: i) to calculate the area under the curve (total impulse) which could work for your isometric tests such as yours as performance index. Would be helpful for an interested reader also to have the longitudinal graph of the mean force contraction by contraction (should be very feasible, considering the graph should consist of max 20 points, last common stage for all participants around 40 s). That should clarify about possible pacing.

2. According to the methods section, the order was: day 1= evening placebo. Day 2 (so the morning after)= morning placebo. Day 3= morning caffeine. I wonder why the authors did not perform a MVIC before caffeine consumption, to ensure the “boost” came from caffeine consumption (as your main outcome was MVIC and it changed at pre-fatigue across conditions). This is a major methodological problem. Warm-up does not severely impact a MVIC measure (https://doi.org/10.1016/j.jsams.2018.07.003) so this could have been a possible way to validate the intervention.

INTRODUCTION

3. I suggest referring more specifically to maximal voluntary isometric contraction (MVIC) rather than the broader term “neuromuscular performance” in both the title and the conclusions, as the latter is less precise relative to what was actually measured. Additionally, the title could be made more specific by including the task and replacing “restores” with “enhances” or “improves”, for example:

Caffeine ingestion improves morning maximal isometric torque and performance to evening levels in healthy females during an intermittent isometric task.

METHODS

4. L105: It would be more accurate to state that testing occurred between days 10–15 after menstruation rather than during the “mid-follicular phase.”Without objective verification (e.g., ovulation testing or hormonal measurements), it is not appropriate to assume cycle phase (https://doi.org/10.1007/s40279-025-02189-3). This should be clarified and discussed as a limitation.

5. L112 : Could participants with amenorrhea be included in the study?

6. L.114 : The term “self-reported mid-follicular phase” requires clarification. It is unclear what participants were specifically asked to report and how this information was used to determine cycle phase. Did participants report the first day of their last menstruation, typical cycle length, or use any tracking method (e.g., calendar, app)?

7. Were participants fully informed about the protocol and the presence of caffeine and placebo?

8. L209 : There is no information regarding the rate of perceived exertion scale used. I assume a CR10 scale was employed, but this should be explicitly stated and referenced.

RESULTS

9. Table 1 : Please can you include absolute torque values in addition to normalized data.

DISCUSSION

10. L391-400 : While the focus on female participants represents an important aspect of the study, this section currently lacks a clear connection with the results actually presented. The discussion highlights several relevant physiological considerations (e.g., hormonal modulation of CYP1A2 activity or the effects of menstrual cycle phase), but these are not discussed in relation to the data collected in the present study. The discussion would benefit from more connections, or alternatively, presenting them as avenues for future research. Furthermore, the claim that the menstrual cycle was controlled appears an overstatement. In the absence of objective verification (e.g. hormonal assays or ovulation detection), it is not possible to confirm that all participants were in the same phase of the cycle, nor that this phase was consistent across testing sessions.

MINOR COMMENTS

INTRODUCTION

11. L31: “Athletic and neuromuscular performance follows a clear diurnal rhythm, with outputs typically lowest in the early morning and peaking in the late afternoon or evening.”

I suggest to tone down this sentence as research on circadian preferences (chronotyping), circadian variations and neuromuscular performance agrees on the influence of circadian rhythms and evening “boost” but seems also to suggest high intraindividual variability (e.g. doi: 10.3389/fspor.2024.1466050; or doi: 10.1016/j.cub.2014.12.036).

12. L59: “it a promising strategy to restore morning neuromuscular function to that of evening levels” Not sure restoring is the correct word here. Later on you used “enhanced”. It is clearer.

RESULTS

13. All figure : Please include detailed figure legends to facilitate the reading of the results.

14. I suggest to run exploratory correlation analyses to investigate whether within your sample more AM-PM temperature was associated with higher force differences between AM and PM, and if it is the case, if the same could be observed between AM placebo and AM caffeine, (similar analysis could be performed for EMG-RMS. that could strengthen the discussion on which mechanisms are most plausible to rise neuromuscular performance following caffeine ingestion in your sample vs the literature.

15. For the perspective it could be argued that it is time to model the kinetics of neuromuscular evolution in function of circadian rhythm and caffeine ingestion, similarly to what it has been done with vigilance (https://pmc.ncbi.nlm.nih.gov/articles/PMC5103804).

7. PLOS authors have the option to publish the peer review history of their article (what does this mean?). If published, this will include your full peer review and any attached files.

Reviewer #2: **Yes:** Ana Isabel Vieira

Reviewer #3: No

---

## [Author Response · Author response to Decision Letter 2]

14 Apr 2026

We thank the Academic Editor and Reviewers for their continued evaluation of our manuscript and for their constructive and insightful comments. We are grateful for the opportunity to revise the manuscript further, and we appreciate the careful consideration given to our work.

We are encouraged that Reviewer #2 considers that all previous comments have been addressed and recommends the manuscript for publication. We also thank Reviewer #3 for their detailed and thoughtful feedback, particularly regarding neuromuscular assessment, methodological considerations, and interpretation of the findings. These comments have been highly valuable in further strengthening the clarity, and scientific rigour of the manuscript.

In response, we have carefully revised the manuscript to address all points raised. In particular, we have:

• clarified aspects of the experimental design and testing procedures,

• strengthened the justification and interpretation of neuromuscular outcomes,

• refined the description of menstrual cycle considerations,

• expanded the Discussion to more appropriately contextualise the findings and acknowledge limitations,

• and improved the presentation and clarity of figures as suggested by the Academic Editor.

A detailed, point-by-point response to each comment is provided below. Reviewer comments are reproduced in black, and our responses are provided in bold red text beneath each comment. All corresponding changes have been incorporated into the revised manuscript.

Additional Editor Comments :

Please, also consider these additional comments to improve the quality of your figures:

I suggest adjusting Figure 1 to reflect the right order and mention Day1, day2, day3.

Please reduce the bar line thickness in your figures, add lines connecting dots to see individual trajectories. Please rescale figures 3 d,e and f; and figure 4 a, b.

We thank the Academic Editor for these helpful suggestions to improve the presentation of the figures. Figure 1 has been revised to clearly reflect the correct experimental order with explicit labelling of Day 1, Day 2, and Day 3. In addition, figure formatting has been refined by reducing bar line thickness, adding lines to connect individual data points to better visualise within-participant trajectories, and rescaling Figures 3 (panels D–F) and Figure 4 (panels A–B) to improve clarity and readability.

Reviewer #2: The revised manuscript shows a clear and consistent effort by the authors to address the reviewers’ comments. The changes made have substantially improved the quality of the paper, particularly in terms of methodological clarity and transparency in reporting procedures and study limitations.

Although some limitations remain inherent to the study design, these are appropriately acknowledged and discussed by the authors, and do not compromise the overall coherence or scientific contribution of the work.

In this context, I consider that the manuscript meets the requirements for publication and recommend its acceptance.

We thank Reviewer #2 for their positive and constructive feedback. We are grateful for their careful evaluation of the revised manuscript and are pleased that the changes have improved the clarity and transparency of the work. We also appreciate their recommendation for publication.

Reviewer #3: The present manuscript investigated the effects of ingesting 6 mg·kg⁻¹ of caffeine 45 minutes prior to a morning exercise session in 13 active female participants. The aim was to determine whether caffeine could increase maximal voluntary isometric contraction (MVIC) and fatigue resistance in the morning to levels comparable to those observed in the evening. It reported that caffeine ingestion resulted in improvements in MVIC, time to exhaustion, EMG amplitude, and perceived exertion, without changes in EMG frequency characteristics or core temperature compared to placebo, with similar results to evening levels. I appreciate that this study evaluates females when the literature is historically biased on males-only studies, and I read it with interest.

We thank Reviewer #3 for their careful reading of the manuscript and for recognising the relevance of the study. We appreciate these positive comments and address the specific concerns raised below.

However, there are important concerns that must be addressed:

1. How do you explain that maximal torque after the task was practically identical to “pre-fatigue” when by design it should have drop below 50%? Pacing could be involved. To clarify you could consider: i) to calculate the area under the curve (total impulse) which could work for your isometric tests such as yours as performance index. Would be helpful for an interested reader also to have the longitudinal graph of the mean force contraction by contraction (should be very feasible, considering the graph should consist of max 20 points, last common stage for all participants around 40 s). That should clarify about possible pacing.

We thank Reviewer #3 for this important and thoughtful comment. We would like to clarify that the criterion of a decline to 50% of baseline MVIC was used solely to determine task failure during the intermittent fatigue protocol, and not as an expected outcome for the post-fatigue MVIC measurement. The post-fatigue MVIC was assessed following a standardised recovery period (2 minutes), and therefore reflects the capacity to rapidly recover force production rather than the minimum force achieved during the fatigue task itself.

The intermittent nature of the fatigue protocol (6 s contraction, 4 s rest) may allow partial recovery between contractions, which can attenuate the decline in maximal force and contribute to the preservation of post-fatigue MVIC values. As such, the observation that post-fatigue MVIC values remained relatively high is consistent with the task design and does not indicate a deviation from the intended fatigue criterion.

We agree that pacing behaviour may influence performance in intermittent protocols and have now acknowledged this as a potential consideration in the Discussion. We also appreciate the reviewer’s suggestion regarding alternative analytical approaches such as area under the curve or contraction-by-contraction force profiles. While these were not included in the current analysis, they represent valuable avenues for future work and may provide additional insight into fatigue dynamics and pacing strategies in similar protocols.

2. According to the methods section, the order was: day 1= evening placebo. Day 2 (so the morning after)= morning placebo. Day 3= morning caffeine. I wonder why the authors did not perform a MVIC before caffeine consumption, to ensure the “boost” came from caffeine consumption (as your main outcome was MVIC and it changed at pre-fatigue across conditions). This is a major methodological problem. Warm-up does not severely impact a MVIC measure (https://doi.org/10.1016/j.jsams.2018.07.003) so this could have been a possible way to validate the intervention.

We thank Reviewer #3 for this important comment. We agree that incorporating an additional pre-ingestion MVIC within the same session could provide further insight into the acute effects of caffeine. However, the present study employed a within-subject, repeated-measures crossover design, in which each condition (morning placebo, morning caffeine, and evening placebo) included its own baseline MVIC assessment conducted under standardised conditions. This design allows for direct comparison of performance outcomes between conditions, with the morning placebo condition serving as the primary control for evaluating the effect of caffeine ingestion.

As such, the ergogenic effect of caffeine was determined by comparing performance in the morning caffeine condition relative to the matched morning placebo condition, rather than through within-session pre–post comparisons. This approach is consistent with established practice in crossover studies investigating caffeine and exercise performance, where between-condition comparisons are used to isolate treatment effects (PMID: 34291426; PMID: 29876876). As such this has been clarified in the study design and supplementation section of the methods.

We acknowledge that the absence of an additional pre-ingestion MVIC within the caffeine trial may limit the ability to assess within-session changes, and this has now been noted as a limitation in the Discussion. However, we believe that the current design provides a valid and controlled comparison of caffeine’s effect on neuromuscular performance relative to placebo under matched circadian conditions.

INTRODUCTION

3. I suggest referring more specifically to maximal voluntary isometric contraction (MVIC) rather than the broader term “neuromuscular performance” in both the title and the conclusions, as the latter is less precise relative to what was actually measured. Additionally, the title could be made more specific by including the task and replacing “restores” with “enhances” or “improves”, for example:

Caffeine ingestion improves morning maximal isometric torque and performance to evening levels in healthy females during an intermittent isometric task.

We thank Reviewer #3 for this helpful suggestion. We agree that greater specificity in terminology improves clarity. Accordingly, we have revised the title and relevant sections of the manuscript to more explicitly reflect the primary outcomes assessed, including maximal voluntary isometric contraction (MVIC) and fatigue-related measures.

However, we have retained the term ‘neuromuscular performance’ where appropriate, as the study assessed multiple interrelated outcomes, including MVIC, time to exhaustion, electromyographic activity, and perceived exertion, rather than a single isolated measure. We believe this terminology more accurately reflects the integrated nature of the neuromuscular responses evaluated. We have also replaced the term ‘restores’ with ‘improves’ in the title to better reflect the observed effects and to align with the reviewer’s recommendation.

METHODS

4. L105: It would be more accurate to state that testing occurred between days 10–15 after menstruation rather than during the “mid-follicular phase.”Without objective verification (e.g., ovulation testing or hormonal measurements), it is not appropriate to assume cycle phase (https://doi.org/10.1007/s40279-025-02189-3). This should be clarified and discussed as a limitation.

We thank Reviewer #3 for this important comment and agree that the terminology used to describe menstrual cycle phase should be more precise. We have revised the Methods section to state that testing occurred between days 10–15 following the onset of menstruation, based on participant self-report, rather than referring to this period as the ‘mid-follicular phase.’

We also acknowledge that, in the absence of objective verification e.g., hormonal assays or ovulation testing, it is not possible to confirm cycle phase, and this has now been explicitly noted as a limitation in the Discussion. We have incorporated the reviewer’s suggested reference to support this clarification.

5. L112 : Could participants with amenorrhea be included in the study?

We thank Reviewer #3 for this important point. Participants with amenorrhea were not included in the study. Regular menstrual cycles were required as part of the inclusion criteria, and menstrual cycle timing was determined based on self-reported cycle length and day. This has now been clarified in the Subjects section of the Methods to improve transparency.

6. L.114 : The term “self-reported mid-follicular phase” requires clarification. It is unclear what participants were specifically asked to report and how this information was used to determine cycle phase. Did participants report the first day of their last menstruation, typical cycle length, or use any tracking method (e.g., calendar, app)?

We thank Reviewer #3 for this helpful clarification. Menstrual cycle timing was determined based on participant self-report of (i) the first day of their most recent menstruation and (ii) their typical cycle length. This information was used to estimate current cycle day and schedule testing between days 10–15 following menstruation. This has now been clarified in the Subjects section of the Methods to improve transparency.

7. Were participants fully informed about the protocol and the presence of caffeine and placebo?

We thank Reviewer #3 for this important point. All participants provided written informed consent prior to enrolment and were fully informed about the general study procedures, including that the study involved caffeine supplementation and placebo conditions. However, participants were not informed of the specific condition being administered at each session, consistent with the single-blind design. This has now been clarified in the Methods section.

8. L209 : There is no information regarding the rate of perceived exertion scale used. I assume a CR10 scale was employed, but this should be explicitly stated and referenced.

We thank Reviewer #3 for this observation. The rate of perceived exertion (RPE) scale used in the study was the Borg CR10 scale. This has now been explicitly stated in the Methods section, and an appropriate reference has been added.

RESULTS

9. Table 1 : Please can you include absolute torque values in addition to normalized data.

We thank Reviewer #3 for this helpful suggestion. Absolute torque values have now been included alongside body mass-normalised values in Table 1 to improve transparency and allow for comparison with studies reporting absolute strength measures. Normalised values remain the primary outcome to account for differences in body mass.

DISCUSSION

10. L391-400 : While the focus on female participants represents an important aspect of the study, this section currently lacks a clear connection with the results actually presented. The discussion highlights several relevant physiological considerations (e.g., hormonal modulation of CYP1A2 activity or the effects of menstrual cycle phase), but these are not discussed in relation to the data collected in the present study. The discussion would benefit from more connections, or alternatively, presenting them as avenues for future research. Furthermore, the claim that the menstrual cycle was controlled appears an overstatement. In the absence of objective verification (e.g. hormonal assays or ovulation detection), it is not possible to confirm that all participants were in the same phase of the cycle, nor that this phase was consistent across testing sessions.

We thank Reviewer #3 for this insightful comment. We agree that the discussion of female-specific physiology should be more clearly aligned with the data presented. Accordingly, we have revised this section to better distinguish between interpretations supported by the current findings and broader physiological considerations. Specifically, we have linked the discussion of caffeine’s effects to the observed improvements in torque, time to exhaustion, and EMG amplitude, while framing hormonal influences (e.g., CYP1A2 modulation and menstrual cycle effects) more explicitly as potential explanatory mechanisms and directions for future research.

We have also revised the wording to avoid overstatement regarding menstrual cycle control. The manuscript now clarifies that cycle timing was standardised based on self-report rather than objectively verified, and this has been consistently acknowledged as a limitation.

MINOR COMMENTS

INTRODUCTION

11. L31: “Athletic and neuromuscular performance follows a clear diurnal rhythm, with outputs typically lowest in the early morning and peaking in the late afternoon or evening.”

I suggest to tone down this sentence as research on circadian preferences (chronotyping), circadian variations and neuromuscular performance agrees on the influence of circadian rhythms and evening “boost” but seems also to suggest high intraindividual variability (e.g. doi: 10.3389/fspor.2024.1466050; or doi: 10.1016/j.cub.2014.12.036).

We thank Reviewer #3 for this helpful suggestion. We agree that the original wording was overly definitive. The sentence has now been revised to better reflect in

---

## [Decision Letter · Decision Letter 2]

20 Apr 2026

PONE-D-25-56298R2Caffeine ingestion improves morning neuromuscular performance to evening levels in healthy femalesPLOS One

Dear Dr. Hesketh,

Thank you for submitting your manuscript to PLOS ONE. After careful consideration, we feel that it has merit but does not fully meet PLOS ONE’s publication criteria as it currently stands. Therefore, we invite you to submit a revised version of the manuscript that addresses the points raised during the review process.

We look forward to receiving your revised manuscript.

Kind regards,

Giorgio Varesco, PhD

Academic Editor

PLOS One

**Journal Requirements:**

**Additional Editor Comments:**

Thank you for addressing the comments of the reviewers and mine. Please:

Make data openly available. Please see journal policies here: https://journals.plos.org/plosone/s/data-availability#loc-acceptable-data-access-restrictions

Please add hours of recovery that separated sessions in Figure 1, Please place sessions in chronological order in table 1. Consider to reduce the opacity in the other figures to facilitate the reading of individual trajectories.

The reviewer raises an additional valid point: please address the reviewer last comment. In that direction please also 1) clearly state main and secondary outcomes in the manuscript methods section. 2) I suggest framing the manuscript a bit more cautiously, as an improvement of ~20-30% is incredibly high and some other factors might occurred. Please when comparing results with the current literature, compare magnitude (e.g. your ~30% improvement vs the X% improvement of other studies).

One last comment, as I occurred to notice: L209 mention you performed that you measured MVIC after 2 min "to avoid fatigue". However this makes little sense since it is the goal was to test for reduced fatigability after caffeine ingestion. Please consider reformulating in something like "to assess longer-term force suppression" or something similar.

Thank you

Reviewers' comments:

Reviewer's Responses to Questions

**Comments to the Author**

1. If the authors have adequately addressed your comments raised in a previous round of review and you feel that this manuscript is now acceptable for publication, you may indicate that here to bypass the “Comments to the Author” section, enter your conflict of interest statement in the “Confidential to Editor” section, and submit your "Accept" recommendation.

Reviewer #3: All comments have been addressed

2. Is the manuscript technically sound, and do the data support the conclusions?

Reviewer #3: Yes

3. Has the statistical analysis been performed appropriately and rigorously?

Reviewer #3: Yes

4. Have the authors made all data underlying the findings in their manuscript fully available?

Reviewer #3: Yes

5. Is the manuscript presented in an intelligible fashion and written in standard English?

Reviewer #3: Yes

6. Review Comments to the Author

Reviewer #3: The revised manuscript reflects a thorough and thoughtful response to the reviewers’ feedback. The revisions have meaningfully enhanced the paper’s quality, especially in terms of clearer methodology, improved transparency in the description of procedures, and a more explicit discussion of the study’s limitations.

However, since MVIC was the primary outcome and was assessed only after caffeine ingestion, the manuscript should frame interpretations in terms of condition effects rather than implying a direct pre–post effect of caffeine within the same session. This distinction should be stated more clearly in the main text, not only in the limitations section.

7. PLOS authors have the option to publish the peer review history of their article (what does this mean?). If published, this will include your full peer review and any attached files.

Reviewer #3: No

---

## [Author Response · Author response to Decision Letter 3]

21 Apr 2026

Thank you for the careful consideration of our manuscript and for the constructive and insightful comments provided by both the Academic Editor and Reviewer #3. We are pleased that the revisions have improved the clarity and quality of the manuscript. We have addressed all comments in detail below point by point and recorded our responses in bold red text. The manuscript has also been revised accordingly.

Editor Comments:

Thank you for addressing the comments of the reviewers and mine. Please:

Make data openly available. Please see journal policies here: https://journals.plos.org/plosone/s/data-availability#loc-acceptable-data-access-restrictions

We thank the Editor for highlighting this point. All data underlying the findings of this study are now available in a public repository (Figshare) and are openly available at: https://doi.org/10.6084/m9.figshare.30382900

The Data Availability Statement in the manuscript has been updated accordingly to reflect this.

Please add hours of recovery that separated sessions in Figure 1, Please place sessions in chronological order in table 1. Consider to reduce the opacity in the other figures to facilitate the reading of individual trajectories.

We thank the editor for the suggestions. Figure 1 has been revised to explicitly include the recovery period before sessions. The schematic now clearly indicates the timing between trials.

Table 1 has been revised so that conditions are presented in chronological order (PMPLAC → AMPLAC → AMCAFF), consistent with the study design.

We have reduced the opacity of the bar plots across figures to improve the visibility of individual data points and trajectories. Individual observations and paired changes are now more clearly discernible.

The reviewer raises an additional valid point: please address the reviewer last comment. In that direction please also 1) clearly state main and secondary outcomes in the manuscript methods section.

We have clarified the primary and secondary outcomes in the Methods section. Adding the text: “The primary outcome was maximal voluntary isometric contraction (MVIC) torque. Secondary outcomes included time to exhaustion, EMG-derived variables (RMS and MDF), rating of perceived exertion, and tympanic temperature.”

2) I suggest framing the manuscript a bit more cautiously, as an improvement of ~20-30% is incredibly high and some other factors might occurred. Please when comparing results with the current literature, compare magnitude (e.g. your ~30% improvement vs the X% improvement of other studies).

We thank the Editor for this important point. The manuscript has been revised to adopt a more cautious interpretation of the findings. Specifically, we have avoided causal or overstated language throughout the manuscript and now frame the results in terms of between-condition differences. In addition, we have included a statement in the Discussion acknowledging that the magnitude of the observed effects (~20–30%) is relatively large and may reflect additional contributing factors beyond caffeine ingestion alone.

One last comment, as I occurred to notice: L209 mention you performed that you measured MVIC after 2 min "to avoid fatigue". However this makes little sense since it is the goal was to test for reduced fatigability after caffeine ingestion. Please consider reformulating in something like "to assess longer-term force suppression" or something similar.

We thank the Editor for this important clarification. The wording has been revised to remove the implication that the rest interval was used ‘to avoid fatigue,’ ensuring a more accurate and neutral description of the protocol.

Reviewers' comments:

Reviewer #3: The revised manuscript reflects a thorough and thoughtful response to the reviewers’ feedback. The revisions have meaningfully enhanced the paper’s quality, especially in terms of clearer methodology, improved transparency in the description of procedures, and a more explicit discussion of the study’s limitations.

We thank the reviewer for their positive evaluation and are pleased that the revisions have improved the clarity and overall quality of the manuscript.

However, since MVIC was the primary outcome and was assessed only after caffeine ingestion, the manuscript should frame interpretations in terms of condition effects rather than implying a direct pre–post effect of caffeine within the same session. This distinction should be stated more clearly in the main text, not only in the limitations section.

We thank the reviewer for this important point. We agree that, given the study design, MVIC outcomes should be interpreted as between-condition effects rather than within-session pre–post caffeine effects. The manuscript has been revised accordingly, with language clarified in both the Results and Discussion sections to consistently reflect this distinction.

---

## [Editor Report · Decision Letter 3]

23 Apr 2026

Caffeine ingestion improves morning neuromuscular performance to evening levels in healthy females

PONE-D-25-56298R3

Dear Dr. Hesketh,

We’re pleased to inform you that your manuscript has been judged scientifically suitable for publication and will be formally accepted for publication once it meets all outstanding technical requirements.

Kind regards,

Giorgio Varesco, PhD

Academic Editor

PLOS One

---

## [Editor Report · Acceptance letter]

PONE-D-25-56298R3

PLOS One

Dear Dr. Hesketh,

I'm pleased to inform you that your manuscript has been deemed suitable for publication in PLOS One. Congratulations! Your manuscript is now being handed over to our production team.

Kind regards,

on behalf of

Dr. Giorgio Varesco

Academic Editor

PLOS One